# NP and 9311 are excellent population parents for screening QTLs of potassium-efficient rice

**Tingchang Liu[1,2], Liangli Bai[3], Lifang Huang[1]\*, Donghai Mao[1]**

**1** Key Laboratory of Agro-Ecological Processes in Subtropical Region, Institute of Subtropical Agriculture, Chinese Academy of Sciences, Changsha, China, **2** University of Chinese Academy of Sciences, Beijing, China, **3** College of Life Sciences, Hunan Normal University, Changsha, China

\* hlf@isa.ac.cn

**Data Availability Statement:** All relevant data are within the manuscript and its Supporting information files.

**Funding:** This work was supported by the Natural Science Foundation of Hunan Province for

## Abstract

High and stable rice yields are critical to global food security, and potassium-deficient soils in East Asia have seriously limited rice production in the regions. It is feasible to screen potassium efficient quantitative trait locus(QTLs) from existing rice varieties to cope with rice production in potassium-deficient areas, and the selection of population parents is the key to locating major QTLs. After a long period of natural selection, potassium efficient rice varieties mainly exist in the region where the soil potassium level is low. The present study chose the representative twelve high-yielding rice varieties in east Asia, firstly, to measure plant height, fresh sheath weight, and fresh blade weight under hydroponic conditions. Based on the difference and consistency of the three parameters, NP as low potassium tolerant, and 9311 as low potassium sensitive rice variety were screened. We further analyzed the relative values of the six parameters of NP and 9311 treated with a culture medium containing different potassium ($K^+$) concentrations and showed that the two varieties significantly differed in multiple low potassium concentrations. Meanwhile, we calculated the coefficient of variation of twelve rice varieties and most of those parameters reached a maximum at 4 mg/L $K^+$, indicating that this concentration was suitable for screening potassium-efficient rice. We also measured the potassium content and the potassium-related traits in NP and 9311 tissues, and found that NP and 9311 significantly differed in potassium translocation. These differences may be responsible for the long-distance transport of potassium from the root to the aboveground part. In conclusion, we identified a pair of parents with significant differences in potassium translocation, which can be used to locate the relevant QTLs with high potassium efficiency to cope with the crisis of soil potassium deficiency in East Asia.

## Introduction

Rice is one of the staple food in the world, feeding more than half the population of the world [1]. Potassium fertilizer plays a vital role in producing a high and stable rice yield. In the East Asia region, potassium fertilizer is scarce, and rice yield is greatly limited by low potassium soil [2]. The method of improving crops has the advantage in terms of economy and environment compared with the method of increasing potassium fertilizer to meet the needs of crops for

Distinguished Young Scholars(2021JJ10041)and Key R&D Programs of Hunan Province (2020WK2023. The funders had no role in study design, data collection and analysis, decision to publish, or preparation of the manuscript.

**Competing interests:** The authors have declared that no competing interests exist.

potassium nutrients. Studies have shown that rice genotypes significantly differ in potassium uptake and utilization [3–6]. It is feasible to alleviate the shortage of potassium resources in these areas by exploring the genetic superiority of superior genotypes through breeding.

The selection of population parents is the key to map potassium-related QTLs. We summarized the current population parents of potassium-related QTLs mapping [7–23] in Table 1. It shows that most of the selected population parents are two varieties that respectively are tolerant and sensitive to salt (potassium is one of them), and there is no obvious differentiation between indica and japonica in the tolerance to salt stress. In general, there are more varieties with the tolerance to salt stress in indica rice. These suggested that selecting a pair of parents with large phenotypic differences under low potassium conditions(salt stress) might be effective when constructing a population to map potassium-related QTLs. The potassium-deficient soil in East Asia has created long-term directional environmental selection pressures for crops, accumulating much natural variability for low potassium tolerance of crops, which provides many options for searching for potassium-efficient rice parents.

In addition, the potassium concentration in the culture medium has a significant influence on screening. When potassium concentration is too low, the advantages of superior genotypes cannot be brought into play. In this case, the differences between genotypes are insignificant and coefficients of variation (CV) are small; when potassium concentration is too high, the selection pressure of potassium is too weak or disappears, which is not conducive to the expression of the genetic potential of different genotypes and to select. An appropriate potassium concentration should not only give full play to the advantages of the varieties with high-efficiency potassium uptake genotypes, but also make the varieties with low-efficiency potassium uptake genotypes have enough selection pressure [24]. It has been proved that screening low potassium tolerance genes in indica and japonica rice by hydroponic liquid with potassium content of 3,10 mg/L $K^+$, and the correlation coefficients of 10 mg /L $K^+$ were highly significant [25]. Eleven rice varieties including indica, japonica and indica hybrid rice, were screened by culture solution with 4,8,12,16 mg/L $K^+$ and the results showed that 8 mg/L $K^+$ solution was suitable for selecting varieties with high potassium uptake genotypes between indica and japonica, and 16 mg/L $K^+$ solution was suitable for screening in hybrid rice [24].

**Table 1. The population parents of QTLs Mapping for traits related to salt tolerance in rice.**

| male parent | response to salt | female parent | response to salt | references |
|---|---|---|---|---|
| IR64 | tolerant Indica | Azucena | sensitive japonica | Wu et al (1998) |
| Pokkali | tolerant indica | IR29 | susceptible indica | BONILLA et al (2002) |
| Nona Bokra | high tolerance indica | Koshihikari | susceptible japonica | Lin et al (2004) |
| Gimbozu | tolerant japonica | Kasalath | susceptible indica | Shimizu et al (2005) |
| Jiucaiqing | strong tolerance japonica | IR36 | sensitive indica | Ming-Zhe et al (2005) |
| Tarommahalli | tolerance indica | Khazar | sensitive indica | Sabouri et al (2008) |
| CSR27 | tolerant indica | MI48 | sensitive indica | Ammar et al (2009) |
| IR61920-3B-22-2-1 (NSIC Rc106) | highly tolerant | BRRI dhan40 | moderately tolerant indica | Islam et al (2011) |
| Tarome-Molaei | tolerant | Tiqing | sensitive | Ahmadi et al (2011) |
| Gharib | tolerant indica | Sepidroud | sensitive indica | Ghomi et al (2013) |
| Changbai10 | tolerant japonica | Dongnong425 | sensitive japonica | Zheng et al (2014) |
| Cheriviruppu | highly tolerant indica | Pusa Basmati 1 (PB1) | highly sensitive indica | Hossain et al (2015) |
| At354 | tolerant indica | Bg352 | susceptible indica | Gimhani et al (2016) |
| Nona Bokra | highly tolerant India | Jupiter | susceptible japonica | Puram et al (2017) |
| Nona Bokra | tolerant indica | Cheniere | susceptible | Puram et al (2018) |
| XieqingzaoB | tolerant indica | Zhonghui9308 (ZH9308) | sensitive indica | Islam et al (2020) |
| XieqingzaoB | tolerant indica | Dongxiang wild rice (O.rufifipogon Griffff.) | sensitive indica | Hu et al (2021) |

Twenty-eight rice varieties were screened with the nutrient solution containing 3, 10, 40 mg/L $K^+$ [26], and the 3 mg/L $K^+$ solution was used to screen out between high potassium tolerance rice and low potassium tolerance rice [27]. Eighteen rice varieties were screened by hydroponic liquid using low potassium (5 mg/L $K^+$) and normal potassium (40 mg/L $K^+$). The results showed that relative plant dry weight, relative stem weight, relative dry root weight, relative root-cap ratio, relative stem leaf potassium uptake, relative plant potassium uptake took a larger proportion in five principal components that can be used as indicators of rice seedling potassium efficient germplasm to screen [28]. About 123 rice double haploid populations, including indica rice IR64, were screened by hydroponic solution with low potassium (5 mg/L $K^+$) and common potassium (40 mg/L $K^+$), and 175 molecular markers distributed on twelve chromosomes were used to analyze QTLs for low potassium tolerance. The results showed that sd-1 had an important effect on plant height under normal and low potassium [29], but four QTLs affecting plant height and three QTLs affecting tiller number were detected under low potassium. In summary, 40 mg/L $K^+$ in hydroponics solution as the normal potassium concentration and 3,4,5,8,10,12,16 mg/L $K^+$ as low potassium concentration has been used to screen potassium-efficient rice in previous studies. Of them, 3 mg/L and 5 mg/L $K^+$ were used repeatedly. Due to multiple varieties having different plant types in the natural growth state, it is unfair to compare the absolute values of each parameter among them. The relative values of each parameter are reasonable.

## Materials and methods

### Plant materials

Wuyunjing, 02428, NP, TB309,TQ,9311, Indica I (J189,J193,J221), and Indica II (J187,J426, J216).

Among them, Wuyunjing is new medium japonica variety with prominent comprehensive traits, which has been actively introduced and promoted for planting due to yield advantages [30]. 02428 is a widely compatible japonica rice variety, which not only has a good affinity to indica rice variety, but also has a strong advantage in the interspecific hybrids configured with it [31]. Nipponbare (Jinyin 153, named as NP) was bred in 1957 by the cross between "Yamahiko" and "Yukufeng" in Aichi Prefecture, Japan. Under the condition of the thin field and low fertilizer cultivation, the yield of this variety is the same or slightly lower than that of "Nonglin 29". It has a higher yield under the culture condition of fruitful and high fat land (https://www.ricedata.cn/variety/varis/602979.HTM).

Taipei 309 (TAI-Pei 309, TP309) has a stable yield under low temperature condition [32]. Rice 93–11 (Mingyang No.6, named as 9311), a medium-mature indica rice variety with high quality, high yield and multiple resistances, has been widely promoted and applied in production [33]. TQ is a new medium rice variety with early maturity, disease resistance and high yield, which was selected and bred by Guangdong Academy of Agricultural Sciences [34]. J189 is a short indica rice variety developed from the combination group of Dijiaowujian (female parent) and Yuanyuan (male parent) [35] in the Agricultural improvement farm of Taichung District. J193 was developed by crossing zhongjiazao 17 (female parent) and G04-44 (male parent), a high yield early indica variety selected from China Rice Research Institute [36]. J221, J187,J426 and J216 are rice varieties widely cultivated in East Asia with high yields.

### Cultural method

All rice seed were treated with distilled water for five days, then treated with 0,1,2,3,4,5,10,20,40,60 mg/L $K^+$ in the hydroponics medium treatment(PH 5.8) for 4 weeks. The nutrient solution was replaced every two days.

## Measuring method

Three replicates were taken from each variety at each treatment, and six plants from each replication at the five-leaf stage. Plant height, root length, root number, fresh root weight, fresh sheath weight and fresh blade weight were measured, and the differences were detected by paired sample T-test with IBM SPSS Statistics 23. Dry root weight, dry sheath weight and dry blade weight were measured after drying, and the potassium content of root, sheath and blade were measured using ICP-OES after grinding. The potassium absorption efficiency, potassium accumulation, potassium absorption rate, the potassium translocation rate from root to sheath, the potassium translocation rate from sheath to blade, potassium distribution rate, potassium utilization efficiency and potassium response index of plant were calculated. Based on normal potassium treatment (40 mg/L K$^+$), the relative values of each treatment group were calculated, and the coefficient of variation (CV% = mean value/ standard deviation) of each measurement parameter at every potassium concentrations were calculated, so as to find out the low potassium concentration of screening different rice varieties. Abbreviations of each investigation parameter are shown in Table 2.

**Table 2. Abbreviations of all investigation parameters.**

| Trait name | Unit | Abbreviation |
|---|---|---|
| Relative potassium absorption efficiency of plant treated with 1 mg/L K$^+$ | | RKAE-1K |
| Relative potassium concentration of plant treated with 1 mg/L K$^+$ | g/100 g | RKC-1K |
| Dry weight of plant treated with 4 mg/L K$^+$ | g | DW-4K |
| Relative dry weight of plant treated with 4 mg/L K$^+$ | | RDW-4K |
| Relative potassium absorption efficiency of plant treated with 4 mg/L K$^+$ | | RKAE-4K |
| Relative potassium accumulation of plant treated with 4 mg/L K$^+$ | | RKA-4K |
| Potassium absorption rate | | KAR |
| Relative potassium translocation rate from root to aboveground part of plant treated with 1 mg/L K$^+$ | | RKT-RTA-1K |
| Potassium translocation rate from root to sheath of plant treated with 1 mg/L K$^+$ | | KT-RTS-1K |
| Relative potassium translocation rate from root to sheath of plant treated with 1 mg/L K$^+$ | | RKT-RTS-1K |
| Relative potassium translocation rate from sheath to blade of plant treated with 1 mg/L K$^+$ | | RKT-STB-1K |
| Relative potassium translocation rate from root to aboveground part of plant treated with 4 mg/L K$^+$ | | RKT-RTA-4K |
| Potassium translocation rate from root to sheath of plant treated with 4 mg/L K$^+$ | | KT-RTS-4K |
| Relative potassium translocation rate from root to sheath of plant treated with 4 mg/L K$^+$ | | RKT-RTS-4K |
| Relative potassium translocation rate from sheath to blade of plant treated with 4 mg/L K$^+$ | | RKT-STB-4K |
| Potassium distribution rate of plant treated with 1 mg/L K$^+$ | | KD-1K |
| Potassium distribution rate of plant treated with 4 mg/L K$^+$ | | KD-4K |
| Relative potassium distribution rate of plant treated with 1 mg/L K$^+$ | | RKD-IK |
| Relative potassium distribution rate of plant treated with 4 mg/L K$^+$ | | RKD-4K |
| Potassium utilization efficiency of plant treated with 1 mg/L K$^+$ | | KUE-1K |
| Potassium utilization efficiency of plant treated with 4 mg/L K$^+$ | | KUE-4K |
| Potassium utilization efficiency of aboveground part of plant treated with 1 mg/L K$^+$ | | KUEA-1K |
| Potassium utilization efficiency of aboveground part of plant treated with 4 mg/L K$^+$ | | KUEA-4K |
| Potassium response index | | KRI |

# Results

## The effects of potassium concentration on plant growth

Plant height is the most obvious trait of cultivated rice, and is also the main index for investigating the growth status of rice. Rice mainly absorbs potassium and other mineral elements through roots, and the uptake amount of various elements determines the biosynthesis amount of aboveground part. Therefore, the biomass of plant aboveground part can be used as a measurement index of growth state. The fresh weight is the tissue mass corresponding to the normal life state of the plant, which reflects the growth state of the plant in the natural state.

In this study, the relative plant height of 12 rice varieties was calculated under different potassium concentration medium treatments (Fig 1; S1 Table). Under the treatment of hydroponic solution with different potassium concentrations, the relative plant height of all rice varieties was gradually decreased with the decrease of potassium concentration in the medium, showing a linear relationship.

Combined with the relative fresh sheath weight (Fig 2; S5 Table) and relative fresh blade weight (Fig 3; S6 Table), there was no obvious indica-japonica differentiation in terms of elative plant height, fresh sheath weight and fresh blade weight of the twelve rice varieties, According to the differentiation and consistency of the three indexes investigated in different varieties under different potassium concentrations of culture medium, NP was initially selected as a low-potassium tolerant variety, while 9311 was more sensitive to a low-potassium culture medium.

## The influence of potassium concentration on the coefficient of variation of the investigated parameter

In order to find out the best potassium concentration in hydroponics medium to distinguish different rice varieties, the coefficients of variation of twelve rice varieties treated with 10 concentrations were calculated (Fig 4; S2–S4 Tables).

The coefficient of variation results reflect the variation range of a certain index of different varieties in medium with a different potassium concentration. The larger the coefficient of

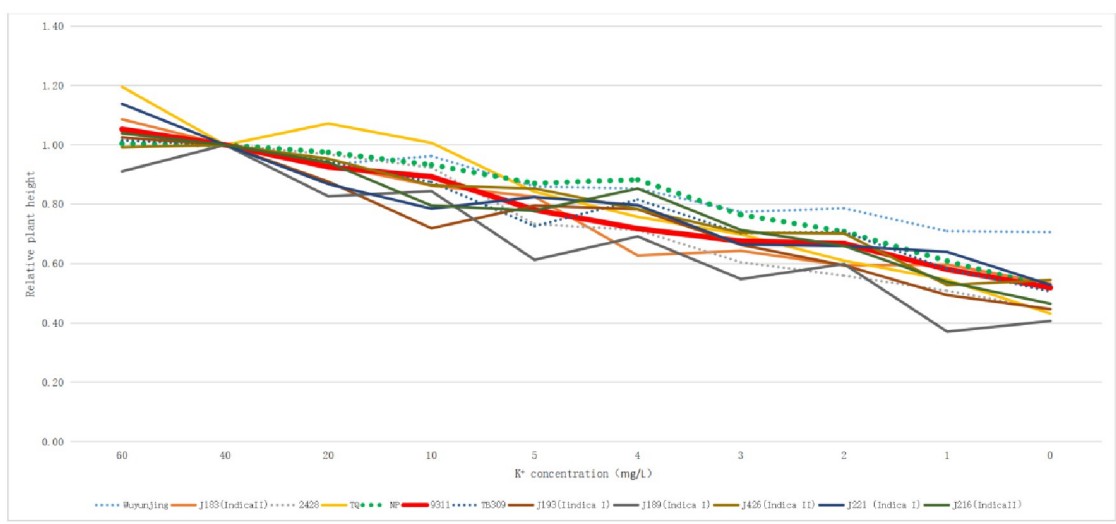

**Fig 1. Relative plant height of twelve rice varieties in different potassium concentration culture solution.** The dotted line is japonica rice and the solid line is indica rice.

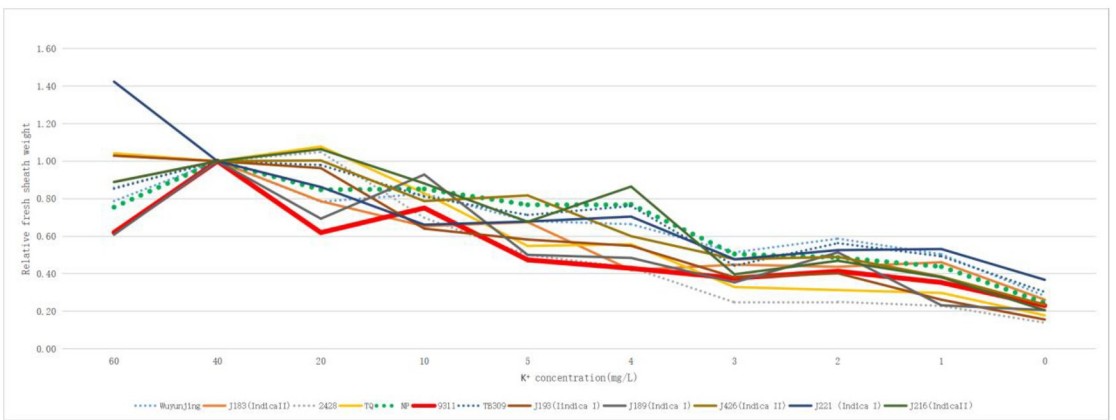

**Fig 2. Relative fresh sheath weight of twelve rice varieties in different potassium concentration culture solution.** The dotted line is japonica rice and the solid line is indica rice.

variation, the larger the variation range of different varieties in a medium with the certain potassium concentration; it is more beneficial for us to distinguish low potassium tolerance among different varieties. When the potassium concentration of nutrient solution for 60 mg/L, all parameter variation coefficients declined precipitously, as the minimum value of different potassium concentration treatments except that in fresh root weight. Increasing the potassium concentration in the culture medium did not increase the coefficient of variation among the cultivars, showing that the potassium concentration has reached its saturation value in twelve rice varieties. As potassium concentration increases, variation range of all varieties decreases, and it is not easy to screen the differences among different varieties. The coefficient of variation for the plant height, root length and fresh sheath weight were higher at 4 mg/L $K^+$ medium treatment. The root number had the highest coefficient variation at 3 mg/L $K^+$ medium treatment, followed by 4 mg/L $K^+$. In conclusion, 4 mg/L $K^+$ was the best concentration for screening the differences among various rice varieties.

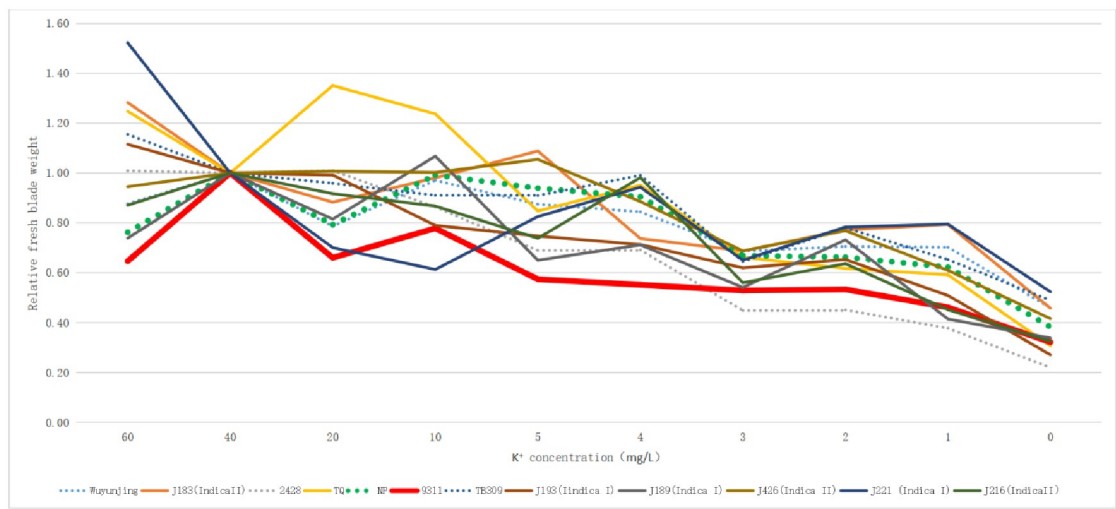

**Fig 3. Relative fresh blade weight of twelve rice varieties in different potassium concentration culture solution.** The dotted line is japonica rice and the solid line is indica rice.

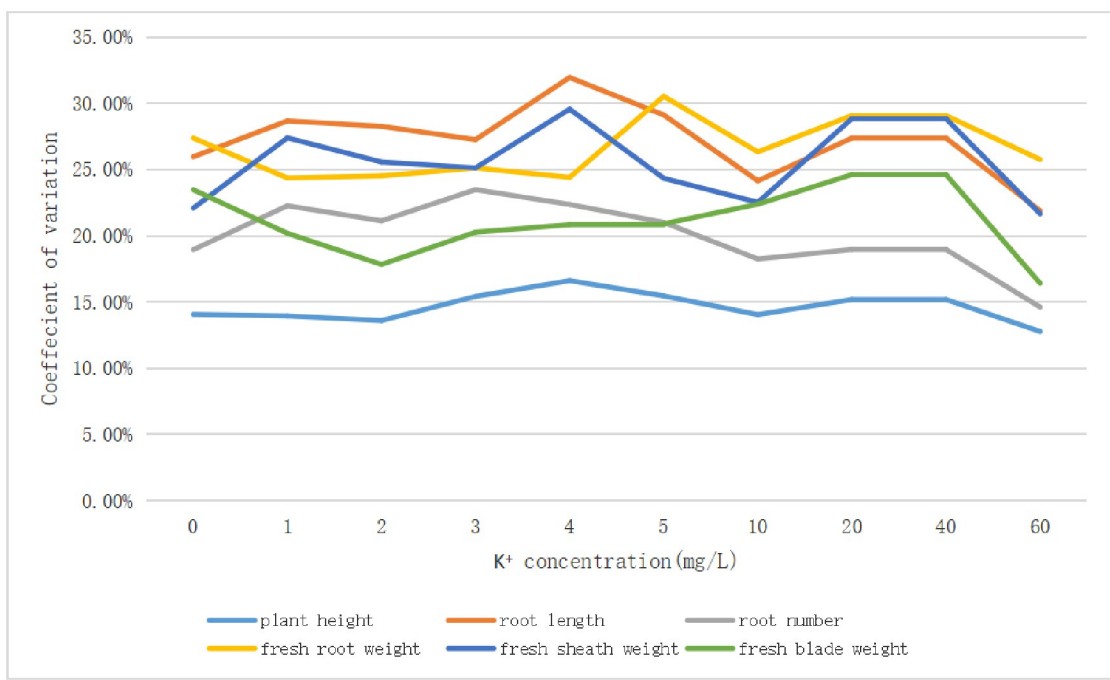

**Fig 4. The variation coefficient of parameters of twelve rice varieties treated with different potassium concentration medium.** The dotted line is japonica rice and the solid line is indica rice.

## Analysis of significant differences between NP and 9311 treated with different potassium concentrations medium

In order to verify the difference between NP and 9311 using 4 mg/ L K$^+$ culture solution, We analyze the significance of six parameters of NP and 9311 after ten potassium concentrations medium treatment (Fig 5; S1–S6 Tables). Two rice varieties were treated with 60 mg/L K$^+$ treatment, and the relative plant height, root length, root number and fresh root weight of 9311 were higher than NP. The relative fresh sheath and fresh blade weight of NP were higher than 9311. Among them, 9311 was significantly higher than NP in relative fresh root weight and root number. NP was significantly higher than 9311 in relative fresh sheath weight and blade weight. When the potassium concentration in the culture medium was lower than 40 mg/L, NP was greater than 9311 in relative plant height, fresh sheath weight and fresh blade weight. When the potassium concentration of culture medium was 4 mg/L, NP was greater than 9311 reached 0.001 level in relative root length and fresh sheath weight, 0.01 level in relative fresh blade weight, and 0.05 level in relative plant height. It indicates that 4 mg/ L K$^+$ treatment can better distinguish between NP and 9311 in the investigated six parameters. Meanwhile, the concentration of 1 mg/L K$^+$ was selected as the extremely low potassium treatment group for subsequent analysis.

## The phenotypes of NP and 9311 were treated with different potassium concentrations

There was no significant difference in appearance between NP and 9311 when treated with 40 mg/L K$^+$ (S1 Fig). However, 9311 showed typical potassium deficiency symptoms with brown and yellow leaf tips on old leaves when treated with 1 mg/L K$^+$ as very low potassium (S2 Fig). 9311 also showed lower plant height than NP when treated with 4 mg/L K$^+$(low potassium)

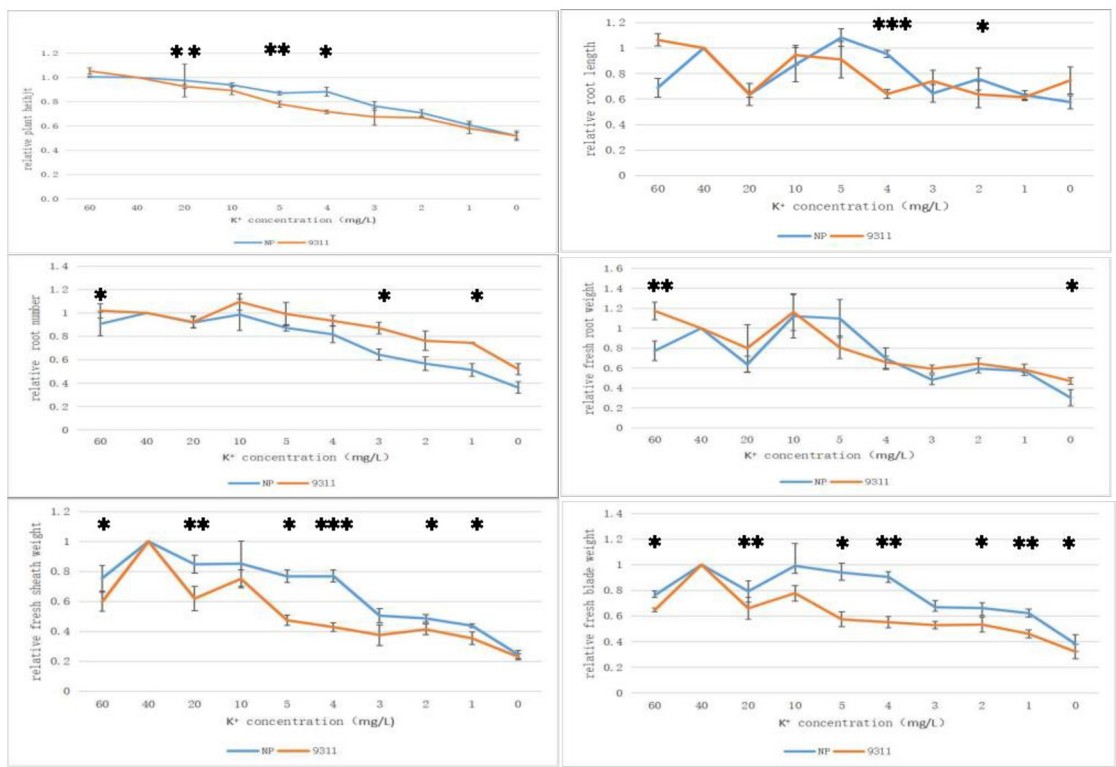

**Fig 5. Significance analysis of six parameters of NP and 9311 in different potassium concentration culture solution.** *:
Significant difference at 0.05 level; **: Very significant difference at 0.01 level; ***: Highly significant difference at 0.001 level.

(Fig 6(a); S3 Fig). There was no significant difference in root length between NP and 9311 at 40 mg/L K⁺ treatment (S4 Fig), but 9311 showed a shorter root length than NP at 4 mg/L K⁺ treatment (S5 Fig). The results indicated that 4 mg/L K⁺ treatment could distinguish NP from 9311 in plant phenotype (Fig 6(b)).

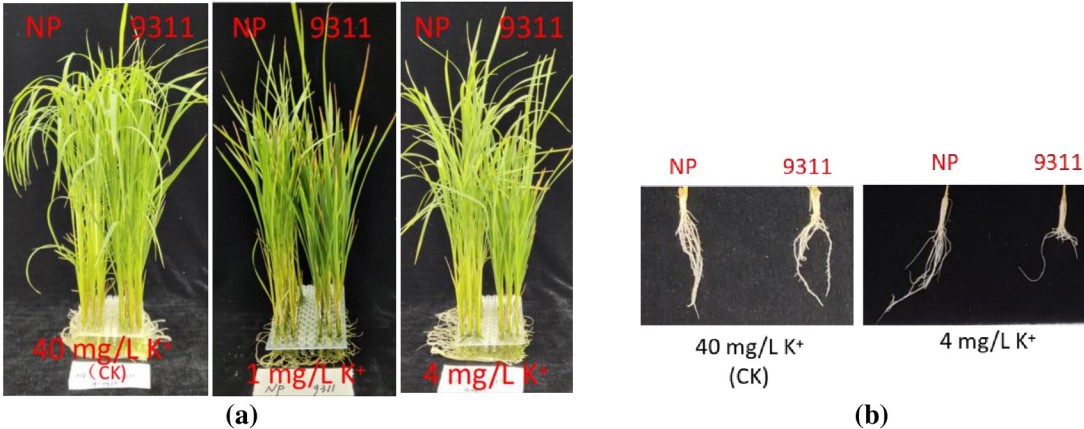

**Fig 6. Phenotypic differences between NP and 9311 treated with low potassium (1, 4 mg/L K⁺) and common potassium (40 mg/L K⁺) medium.** a: Plant growth state; b: Root state of the plant.

### The differences of potassium-related traits between NP and 9311 treated with low potassium medium

Potassium efficiency in rice physiological characteristics embodied in the roots potassium uptake efficiency from environment is high, potassium translocation rate from roots to aboveground part is large, transfer speed is quick, and high potassium accumulation in plant [37]. The photosynthesis of the sheath and blade mainly provides an energy source that late rice growth requires. So potassium efficient rice requires that potassium absorbed by root and is distributed more ratio in leaf. When the same amount of potassium is absorbed from the outside environment, potassium efficient rice must increase utilization efficiency for growth and development. Conversely, when exposed to a low potassium medium environment, potassium efficient rice must respond quickly to low potassium signals to reduce the adverse effects. Multiple parameters of NP and 9311 were analyzed under three potassium concentrations (Table 3; S7–S9 Tables). Compared with 9311, NP had a greater potassium response index when treated with low potassium nutrient solution, including the potassium absorption rate and the potassium translocation rate from root to sheath. Likewise, the potassium accumulation concentration, the potassium translocation rate from root to shoot (including the potassium translocation rate from root to sheath and from sheath to blade), and the potassium distribution rate of aboveground part relative to the common potassium concentration (40 mg/ L $K^+$) were higher in NP as compared with 9311. There was no significant difference in potassium utilization efficiency between NP and 9311 treated with a low potassium medium. However NP had a higher relative potassium accumulation and the potassium utilization efficiency of the plant aboveground part compared with 9311.

## Discussion

Previous studies showed no obvious differentiation of this trait between indica and japonica when rice was screened for salt stress tolerance (including low potassium tolerance) (Table 1). The phenotypes of relative plant height, fresh sheath weight and fresh leaf weight in the present study were consistent with those of previous studies. Previous studies have shown more varieties with the tolerance to low potassium tolerance in indica rice. In this paper, twelve representative rice parents in East Asia were screened, NP as a tolerant japonica, and 9311 as a sensitive indica, which was a pair of rare population parents than previous studies. By constructing the two parents, it was possible to locate the QTLs with potassium-efficient different from previous studies. In previous studies, 3 mg/L and 5 mg/L $K^+$ hydroponic solution were repeatedly used as low potassium screening concentrations [25–29]. The study showed that most of those coefficient of variation reached a maximum at 4 mg/L $K^+$ medium treatment among the investigated six parameters, indicating that this concentration may be a better concentration for screening potassium efficient rice, and as confirmed by the further analysis of NP and 9311 using six parameters.

When treated with an extremely low potassium hydroponics medium, the relative potassium concentration in NP was the same as that in 9311, and the potassium efficiency of NP was mainly manifested in dry matter accumulation of plant. The higher potassium translocation rate from root to the aboveground part of NP was further analyzed as the higher potassium translocation rate from root to sheath of NP, indicating that those genes differential expression mainly caused the difference in potassium translocation between NP and 9311 were related to potassium in long-distance transport, which is responsible for transporting potassium absorbed by root to the aboveground part. According to the classification system of transporters (Transporters Classification, TC), Potassium Transporters [38] can be divided into channel/hole transporter protein and electrochemical potential driver, these two transporters mediate two kinetic potassium uptake processes in plant [39]. Among them, the second type of transporters

**Table 3. Difference analysis of potassium-related traits of NP and 9311 treated with low potassium medium.**

| Trait name | NP | 9311 |
| --- | --- | --- |
| RKAE-1K[1,2] | 4.4768 | 4.4065 |
| RKC-1K[1] | 0.1119 | 0.1102 |
| DW-4K | 0.0479 | 0.0408 |
| RDW-4K[1] | 0.9050 | 0.6428 |
| RKAE-4K[1,2] | 2.2073 | 2.0985 |
| RKA-4K[1,3] | 0.1998 | 0.1349 |
| KAR[4] | 0.1742 | 0.1328 |
| RKT-RTA-1K[1,5] | 0.0141 | 0.0112 |
| KT-RTS-1K[6] | 0.3773 | 0.2493 |
| RKT-RTS-1K[1,6] | 0.0119 | 0.0087 |
| RKT-STB-1K[1,7] | 0.0170 | 0.0167 |
| RKT-RTA-4K[1,5] | 0.0412 | 0.0306 |
| KT-RTS-4K[6] | 1.1930 | 0.7086 |
| RKT-RTS-4K[1,6] | 0.0376 | 0.0247 |
| RKT-STB-4K[1,7] | 0.0608 | 0.0734 |
| KD-1K[8] | 0.8498 | 0.8244 |
| KD-4K[8] | 0.9153 | 0.8881 |
| RKD-IK[1,8] | 0.9493 | 0.9376 |
| RKD-4K[1,8] | 1.0225 | 1.0099 |
| KUE-1K[9] | 200.6569 | 198.4192 |
| KUE-4K[9] | 101.7436 | 104.1616 |
| KUEA-1K[10] | 1.5915 | 1.5058 |
| KUEA-4K[10] | 0.8282 | 0.8014 |
| KRI[11] | 0.0059 | 0.0016 |

Note: 1:This value is relative to the value of 40 mg/L $K^+$ treatment; 2:Relative potassium absorption efficiency = average plant potassium concentration/culture medium potassium concentration; 3:Relative potassium accumulation = relative potassium mass of plant treated with 4 mg/L $K^+$; 4:Potassium absorption rate = (potassium content of plant treated with 40 mg/L $K^+$—potassium content of plant treated with 1 mg/L $K^+$) /[100*28*(dry root weight of plant treated with 40 mg/L $K^+$ + dry root weight of plant treated with 1 mg/L $K^+$)/2]; 5:Potassium translocation rate from root to aboveground part of plant = potassium concentration in aboveground part/potassium concentration in root; 6:Potassium translocation rate from root to sheath of plant = potassium concentration in sheath/potassium concentration in root; 7:Potassium translocation rate from sheath to blade of plant = potassium concentration of blade /potassium concentration of sheath; 8:Potassium distribution rate = potassium mass of in aboveground part/potassium mass in whole plant; 9:Potassium utilization efficiency = total dry weight of plant/total potassium mass of plant; 10:Potassium utilization efficiency of plant aboveground part = total dry weight of plant aboveground part / total potassium mass of plant; 11:Potassium response index = (total dry weight of plant treated with 4 mg/L $K^+$—total dry weight of plant treated with 1 mg/L $K^+$)/difference value of potassium supply concentration.

mainly belongs to transporters (TC: 2.A), including unidirectional transporters, co-transporters and anti-transporters. According to their functional and structural properties, these potassium transporters have been divided into three potassium channel families (Shaker family, TPK family and Kir-like family) and three transporter families (KUP/HAK/KT family, HKT family, and CPA family) [40–42]. SKOR1, the outward potassium transporter of the Shaker family, is mainly expressed in xylem parenchyma cells and transfers potassium absorbed by root cells to xylem cells. In order to maintain the charge balance of xylem wound fluid, potassium ions are often co-transported with other anions during long-distance transport. Except

for TPK4 is localized to cytoplasmic membrane and endoplasmic reticulum, all other TPK family members are localized to the vacuolar membrane [43–47], which may be involved in the regulation of intracellular potassium balance and cellular osmotic potential. Kir-like potassium channels have been found only in Arabidopsis, and only one member, KCO3, has been found [46]. KCO3 exists as a dimer in vivo and is localized on vacuolar membranes. KCO3 mutants are inhibited in growth under osmotic stress, suggesting that KCO3 may be involved in plant osmoregulation [48]. HKT in rice is divided into two families, including eight OsHKT1 members, mainly expressed in the root xylem companion cells and phloem, and it has no potassium transport activity, just as the $Na^+$ one-way transporter, involved in plant of long-distance transport of $Na^+$, by transporting $Na^+$ in the crown to root, the decrease of $Na^+$ content in the xylem bleeding sap, reduce the $Na^+$ accumulation of crown, Increased salt tolerance of plants [49–51]. It was found that multiple members of the first subfamily of HKT were involved in the QTLs of plant salt tolerance using $Na^+/K^+$ ratio, indicating that members were involved in the process of plant response to salt stress by regulating $Na^+/K^+$ ratio in plants [52, 53]. The HKT in rice OsHKT2;1 is mainly expressed in the epidermis, cortex and endodermis of root, and can permeate potassium ions, and its selectivity for $Na^+$ and $K^+$ depends on the external $Na^+$ and $K^+$ concentration [54, 55], but there is no experimental evidence of its involvement in plant potassium nutrition, which may absorb $Na^+$ to compensate for the lack of $K^+$ osmosis regulatory function under low potassium conditions [56]. OsHKT2;4 is mainly expressed in epidermal cells and has high selectivity for potassium ions [57, 58], which may be involved in the process of potassium uptake by plant roots. KUP/HAK/KT in rice can be divided into four subfamilies. The first subfamily members encode high-affinity potassium transporters. It may be mainly involved in the translocation of high-affinity potassium in plants under low potassium conditions [59, 60]. The functions of the second subfamily are diverse, but their involvement in plant potassium uptake remains unclear. This family mediates low-affinity potassium translocation and may act in coordination with potassium channels. Nitrate Transporter 1/Peptide Transporter Family (NPF) plays an important role in the xylem cells of rice. KT/HAK/KUP transporters (including AtKUP7 and OsHAK5) and NPF play a key role in potassium translocation from root to stem [61]. It can be speculated that NP is more resistant to low potassium than 9311, which may be caused by the differential expression of genes controlling the synthesis of SKOR1, NPF, KT/HAK/KUP transporters protein family.

## Conclusion

Based on phenotypic screening and measurement of potassium content, NP as a low potassium tolerance and 9311 as a low potassium sensitive variety were selected from twelve varieties representing East Asia. According to the phenotype, treated with a low potassium hydroponics medium can clearly distinguish the difference between NP and 9311, multiple agronomic traits reach significant differences when treated with 4 mg/ L $K^+$ hydroponics medium. The NP and 9311 varieties could also be distinguished by potassium-related traits, which confirmed that the two varieties had marked differences in potassium translocation. The QTLs with high potassium efficiency could be located by constructing the recombinant inbred line or introgression line of the two varieties.

## Supporting information

**S1 Fig. Phenotypic differences between NP and 9311 treated with 40 mg.L-1 potassium medium.**
(TIF)

**S2 Fig. Phenotypic differences between NP and 9311 treated with 1 mg.L-1 potassium medium.**
(TIF)

**S3 Fig. Phenotypic differences between NP and 9311 treated with 4 mg.L-1 potassium medium.**
(TIF)

**S4 Fig. Root phenotypic differences between NP and 9311 treated with 40 mg.L-1 potassium medium.**
(TIF)

**S5 Fig. Root phenotypic differences between NP and 9311 treated with 4 mg.L-1 potassium medium.**
(TIF)

**S1 Table. Plant height of twelve rice varieties in different potassium concentration culture solution.**
(ODT)

**S2 Table. Root length of twelve rice varieties in different potassium concentration culture solution.**
(ODT)

**S3 Table. Root number of twelve rice varieties in different potassium concentration culture solution.**
(ODT)

**S4 Table. Fresh root weight of twelve rice varieties in different potassium concentration culture solution.**
(ODT)

**S5 Table. Fresh sheath weight of twelve rice varieties in different potassium concentration culture solution.**
(ODT)

**S6 Table. Fresh blade weight of twelve rice varieties in different potassium concentration culture solution.**
(ODT)

**S7 Table. Potassium content of twelve rice varieties in 1 mg.L-1 potassium culture solution.**
(ODT)

**S8 Table. Potassium content of twelve rice varieties in 4 mg.L-1 potassium culture solution.**
(ODT)

**S9 Table. Potassium content of twelve rice varieties in 40 mg.L-1 potassium culture solution.**
(ODT)

## Acknowledgments

The authors would like to thank Dr. Kefyalew for his helpful revise to this manuscript.

## Author Contributions

**Conceptualization:** Donghai Mao.

**Data curation:** Tingchang Liu.

**Funding acquisition:** Lifang Huang.

**Methodology:** Tingchang Liu, Liangli Bai.

**Writing – original draft:** Tingchang Liu.

**Writing – review & editing:** Tingchang Liu.

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
