## [Decision Letter · Decision Letter 0]

14 Mar 2023

PONE-D-22-24710NP and 9311 are excellent population parents for screening QTLs of potassium-efficient ricePLOS ONE

Dear Dr. Lifang Huang,

Thank you for submitting your manuscript to PLOS ONE. After careful consideration, we feel that it has merit but does not fully meet PLOS ONE’s publication criteria as it currently stands. Therefore, we invite you to submit a revised version of the manuscript that addresses the points raised during the review process.

We look forward to receiving your revised manuscript.

Kind regards,

Rohit Joshi, Ph.D.

Academic Editor

PLOS ONE

Journal Requirements:

“This work was supported by the Natural Science Foundation of Hunan Province for Distinguished Young Scholars(2021JJ10041)and Key R&D Programs of Hunan Province(2020WK2023 .The funders had no role in study design, data collection and analysis, decision to publish, or preparation of the manuscript.”

3. PLOS requires an ORCID iD for the corresponding author in Editorial Manager on papers submitted after December 6th, 2016. Please ensure that you have an ORCID iD and that it is validated in Editorial Manager. To do this, go to ‘Update my Information’ (in the upper left-hand corner of the main menu), and click on the Fetch/Validate link next to the ORCID field. This will take you to the ORCID site and allow you to create a new iD or authenticate a pre-existing iD in Editorial Manager. Please see the following video for instructions on linking an ORCID iD to your Editorial Manager account: https://www.youtube.com/watch?v=_xcclfuvtxQ.

Reviewers' comments:

Reviewer's Responses to Questions

**Comments to the Author**

1. Is the manuscript technically sound, and do the data support the conclusions?

Reviewer #1: Yes

2. Has the statistical analysis been performed appropriately and rigorously? 

Reviewer #1: Yes

3. Have the authors made all data underlying the findings in their manuscript fully available?

Reviewer #1: Yes

4. Is the manuscript presented in an intelligible fashion and written in standard English?

Reviewer #1: Yes

5. Review Comments to the Author

Reviewer #1: This manuscript by Liu et al. tested twelve high-yielding rice varieties for K efficiency under hydroponic conditions. The NP and 9311 were identified as tolerant and sensitive varieties in low potassium medium and recommended as potential parents for QTL screening based on their contrasting features. The paper consists of adequate originality, scientific quality, relevance to the field of this journal and presentation; therefore, it is suitable to publish after minor revision. My suggestion is as follows:

1. In the title of the manuscript, the authors mentioned NP and 9311 are excellent population parents for screening QTLs of K-efficient rice. Since I haven't seen any breakthrough in the QTL analysis, so I was very excited when I read the paper; however, I did not find any outcome regarding the QTLs identification. The recommendation was based only on the contrasting features of the NP and 9311 under variable K concentration.

2. In the plant materials section, the authors may incorporate some background details of the used varieties, such as origin, year of release, recommended soil type and zone for different varieties.

3. Check the spelling “lever” or “level” in the abstract.

4. Some references are missing or irregular in the text. In table 1, the author cited Ghomi et al. (2013) and YAO et al. (2005) but not indexed in the reference section. Likewise, the citation of the reference Ming-zhe et al. (2005) (lines 302-304) is missing in the manuscript’s text. Authors must follow the journal format for reference writing and could use the PLOS One template to prepare.

6. PLOS authors have the option to publish the peer review history of their article (what does this mean?). If published, this will include your full peer review and any attached files.

Reviewer #1: **Yes: **Sarfraz Ahmad

---

## [Author Response · Author response to Decision Letter 0]

20 Mar 2023

Dear Editor:

We do appreciate the time and effort you and the reviewers dedicated to providing feedback on our manuscript and are grateful for the insightful comments on our paper. We have tried to address all the raised concerns accordingly. All changes are highlighted. Here is a point-by-point response for all comments and concerns.

Reviewer #1

This manuscript by Liu et al. tested twelve high-yielding rice varieties for K efficiency under hydroponic conditions. The NP and 9311 were identified as tolerant and sensitive varieties in low potassium medium and recommended as potential parents for QTL screening based on their contrasting features. The paper consists of adequate originality, scientific quality, relevance to the field of this journal and presentation; therefore, it is suitable to publish after minor revision. My suggestion is as follows:

Question 1: In the title of the manuscript, the authors mentioned NP and 9311 are excellent population parents for screening QTLs of K-efficient rice. Since I haven't seen any breakthrough in the QTL analysis, so I was very excited when I read the paper; however, I did not find any outcome regarding the QTLs identification. The recommendation was based only on the contrasting features of the NP and 9311 under variable K concentration.

Response: Population parents with significant phenotypic differences under low potassium levels were screened from 12 high-yield rice varieties in East Asia in this study, and a scientific and reliable method for phenotypic screening was established, which could publish as a phased result. As we know, exploring new genes in plant potassium-efficient and analyzing its mechanism are hot topics in current research, and different research teams are very competitive in this field. Based on the parents and screening methods established in our lab, we use NP as the donor parent and 9311 as the recipient parent to construct the introversion line population, and we will use the established method to identify the low potassium tolerance of the progeny lines. As the selected parents are rare combinations, it is very promising to locate the QTLs sites different from those of previous generations. We hope that later research results will be published in the journal.

Question 2: In the plant materials section, the authors may incorporate some background details of the used varieties, such as origin, year of release, recommended soil type and zone for different varieties.

Response: We have rewritten the Plant materials section and added the information about the 12 rice varieties used in this paper， such as origin, year of release, recommended soil type and zone for different varieties as marked with blue color. Related to this, we add the following references as marked with green color:

30. Jianhua L, Zhihai Y, Zhongyi N, Maosong H. Characteristics and key cultivation techniques of Wuyuning 8. ChinaRice. 1999;3(15).

31. Ziming W, Jiangshi Z, Weixiang Z, Chengkuan C, Guiyuan Z, Xiaolu S. Breeding of new wide compatibility strains of japonica rice. Selective Breeding. 1990:32-6. https://doi.org/10.16267/j.cnki.1005-3956

32. Chun-xiu S, Zi-lei Y, Zhi-qun Q. Physiological and Morphological Responses of the Rice TP309 to Short － term Cadmium Stress. Journal of Yichun University. 2019;41(12):5-9.

34. Gui-mao Y. A new medium rice variety Teqing No. 1 with high yield and disease resistance was successfully planted. Technology in Brief 1995:33. https://doi.org/10.14088/j.cnki.issn0439.8114

35. Shicheng L, Shao-kai M. Rice varieties and their pedigrees in China. Shanghai Science and Technology Press. 1991;(24). 

36. Xiaonian W, Qiuping Y, Changming X, Chuanyong W, Linli Z. Breeding and cultivation techniques of a new two-line hybrid early rice combination 'Lingliangyou 193'. South China Agriculture. 2015;9(6):1-2. https://doi.org/10.19415/j.cnki.1673-890x

Question 3: Check the spelling “lever” or “level” in the abstract.

response: We sincerely thank the reviewer for careful reading. As suggested by the reviewer, we have corrected the “lever” into “level” in revised version of the manuscript as marked with red color.

Question 4: Some references are missing or irregular in the text. In table 1, the author cited Ghomi et al. (2013) and YAO et al. (2005) but not indexed in the reference section. Likewise, the citation of the reference Ming-zhe et al. (2005) (lines 302-304) is missing in the manuscript’s text. Authors must follow the journal format for reference writing and could use the PLOS One template to prepare.

Response: Thank you for pointing this out. We have changed the citation format of YAO et al. (2005) and Ming-zhe et al. (2005) (lines 302-304) as marked with purple color, and added the citation as marked with yellow color:

11. Ghomi K, Rabiei B, Sabouri H, Sabouri A. Mapping QTLs for traits related to salinity tolerance at seedling stage of rice (Oryza sativa L.): an agrigenomics study of an Iranian rice population. OMICS. 2013;17(5):242-51.https://doi.org/10.1089/omi.2012.0097 PMID:23638881

---

## [Decision Letter · Decision Letter 1]

3 Apr 2023

NP and 9311 are excellent population parents for screening QTLs of potassium-efficient rice

PONE-D-22-24710R1

Dear Dr. Lifang Huang,

We’re pleased to inform you that your manuscript has been judged scientifically suitable for publication and will be formally accepted for publication once it meets all outstanding technical requirements.

Kind regards,

Rohit Joshi, Ph.D.

Academic Editor

PLOS ONE

Additional Editor Comments (optional):

Reviewers' comments:

Reviewer's Responses to Questions

**Comments to the Author**

1. If the authors have adequately addressed your comments raised in a previous round of review and you feel that this manuscript is now acceptable for publication, you may indicate that here to bypass the “Comments to the Author” section, enter your conflict of interest statement in the “Confidential to Editor” section, and submit your "Accept" recommendation.

Reviewer #1: All comments have been addressed

2. Is the manuscript technically sound, and do the data support the conclusions?

Reviewer #1: Yes

3. Has the statistical analysis been performed appropriately and rigorously? 

Reviewer #1: Yes

4. Have the authors made all data underlying the findings in their manuscript fully available?

Reviewer #1: Yes

5. Is the manuscript presented in an intelligible fashion and written in standard English?

Reviewer #1: Yes

6. Review Comments to the Author

Reviewer #1: (No Response)

7. PLOS authors have the option to publish the peer review history of their article (what does this mean?). If published, this will include your full peer review and any attached files.

Reviewer #1: **Yes: **Sarfraz Ahmad

---

## [Editor Report · Acceptance letter]

6 Apr 2023

PONE-D-22-24710R1 

NP and 9311 are excellent population parents for screening QTLs of potassium-efficient rice 

Dear Dr. Huang:

I'm pleased to inform you that your manuscript has been deemed suitable for publication in PLOS ONE. Congratulations! Your manuscript is now with our production department. 

Kind regards, 

on behalf of

Dr. Rohit Joshi 

Academic Editor

PLOS ONE